# Land Use Improvement as the Drought Mitigation to Manage Climate Change in the Dodokan Watershed, Lombok, Indonesia

Ryke Nandini [1] and Ambar Kusumandari [2,*]

1   Research Center for Ecology and Ethnobiology, National Research and Innovation Agency, Jl. Raya Jakarta-Bogor KM 46, Cibinong 16911, Indonesia; ryke.nandini@brin.go.id
2   Faculty of Forestry, Universitas Gadjah Mada, Jl. Agro No. 1, Yogyakarta 55281, Indonesia
*   Correspondence: ambar_kusumandari@ugm.ac.id; Tel.: +62-82138856645

**Abstract:** The Dodokan watershed is one of the priority watersheds on Lombok Island, which is a mainstay for providing water for the community but has a high vulnerability to drought. This condition causes disruption to the fulfillment of water needs, so it needs to be anticipated early with mitigation actions. Understanding the trend of drought is needed as a basis for developing mitigation measures, especially related to land use in the watershed. This study aims to analyze the hydro-meteorological drought trend that occurred from 2009 to 2020 and to find out the role of land use improvement in mitigating drought, which is predicted in the next 10 years. The Thornthwaite–Mather method was used for drought analysis, and its predictions for the next 10 years were determined by using trend analysis. Land use simulation using a spatial analysis was carried out as an effort to prepare for drought mitigation actions. The results show that the drought in the Dodokan Watershed has increased in the period between 2009–2020 and is predicted to continue to increase. Land use improvement by restoring the forest is a more appropriate mitigation method to overcome the drought than rearranging the agricultural land on slopes above 25%. The simulation results have succeeded in increasing the water surplus and reducing the water deficit in 2030 so that it can be used for drought mitigation in the Dodokan watershed.

**Keywords:** drought index; drought mitigation; climate change; water balance; Thornthwaite–Mather; watershed

## 1. Introduction

Drought is an issue that commonly affects several areas of the Lombok Island [1]. One of the characteristics of the climate in Lombok is the low rainfall and the short number of rainy days, making Lombok vulnerable to drought [2]. The average rainfall in West Nusa Tenggara is in the range of <2000 mm/y over the last 160 days [3,4]. The National Board for Disaster Management stated that there were at least 369 drought-prone villages in the medium category and 15 villages within the high drought vulnerability category [5]. The impact of the drought on Lombok was felt by various sectors. As a small island, the Lombok Island consists of 197 watersheds, of which, according to WWF Nusa Tenggara, more than 70% are critical watersheds [6]. The Dodokan Watershed is one of the priority watersheds determined by the Decree of the Minister of Forestry and Plantations No. 284/KPTS-II/1999. Floods and droughts are the main problems often faced by the Dodokan Watershed. This is in accordance with the character of the Dodokan Watershed, where in the rainy season there will be flooding and in the dry season it will experience drought.

With rainfall that is included in the very low to low categories and climate types C, D, and E according to Schmidt–Ferguson [7], drought is increasingly threatening the role of the Dodokan Watershed in providing water sources for the community, especially in the West Lombok and Central Lombok regencies, which rely on meeting their water needs from the Dodokan Watershed. Ref. [8] stated that during the period of 2012–2017, the

Dodokan Watershed began to experience hydrological drought regularly. The Dodokan Moyosari Watershed Management Institute stated that in 2019, the water use index value in the Dodokan Watershed was within the very poor category, which could be interpreted as prone to water shortages. In West Nusa Tenggara, there was an increase in the number of residents affected by drought by 102,609 people in 2017–2020 [9]. Along with the increase in population, it is estimated that shortages will increase, especially with the phenomenon of climate change as stated by [10].

The drought that occurred on Lombok Island was caused by the influence of rainfall (meteorological drought) which had an impact on hydrological and agricultural droughts. Refs. [11,12] mentioned that the meteorological drought, which is characterized by the frequency and duration of drought, causes a hydrological drought, which is indicated by a water deficit, among others. However, both meteorological and hydrological droughts do not always have a linear relationship, as stated by [13]. Much research on drought on Lombok Island has been carried out, including in Ref. [2], which analyzed the drought on Lombok Island during 1996–2015; in Ref. [14], which analyzed the hydrological drought in Central Lombok in 1995–2018; and also in Ref. [1], which analyzed the relationship between meteorological and agricultural drought on Lombok Island in the period of 2001–2018. Various works of research state that drought is not only influenced by meteorological factors, but also by physical and anthropogenic factors [15–18]. Drought is a problem that has a complex impact on many sectors of life, so it cannot be managed partially [19]. In the West Nusa Tenggara Province, one of the efforts to meet the water needs as a result of the drought is to provide additional water supplies through water tanks which are distributed to the affected areas. The Regional Disaster Management Agency of the West Nusa Tenggara Province said that in 2018, more than 885,600 L of water were distributed throughout Lombok Island [9].

As for dealing with agricultural drought, adaptation strategies have been carried out by the community, among others, by regulating cropping patterns and selecting plant species that are adaptive to drought [20]. Drought is closely related to water availability, whose availability is decreasing along with the phenomenon of climate change as well as an increase in the amount of water demand [21,22]. Several studies have stated that water availability is closely related to land use, one of which is the adequacy of the forest area in a watershed [23,24]. The increasing needs of the population cause pressure on land to increase as well, and have an impact on land conversion. Land use changes including deforestation and afforestation have a significant influence on the occurrence of floods and droughts in a watershed [25,26]. In the Dodokan Watershed, deforestation has also occurred. Based on data released by the Ministry of Environment and Forestry (MoEF) of the Indonesian Republic, in 2009–2020 there was a significant decline in the area of primary forest in the Dodokan Watershed, which is around 86%, while secondary forest has decreased by 27%. The decline in forest area was partly due to the conversion of forest area functions in the form of other land uses such as rice fields, dry land agriculture, mixed dry land agriculture, shrubs, and even settlements. Ref. [8] stated that the conversion of forest areas in the Dodokan Watershed caused changes in hydrological conditions, including the frequency of floods and droughts. Thus, it is necessary to make land use improvements to restore the hydrological function of the Dodokan Watershed.

Climate change, which has been occurring for several decades, will, in the long-term, affect many aspects of life, including the fulfillment of water needs, which will cause a water crisis [22]. The fulfillment of water needs is one indicator of achieving sustainable development goals (SDGs), which Indonesia targets to be globally achieved by 2030. Currently, the West Nusa Tenggara Provincial Government has a guide to action plans for climate change mitigation, as stated in the West Nusa Tenggara Governor's Regulation No. 51 of 2012. Many mitigation strategies are chosen as a means of overcoming various disasters, particularly those related to climate change because they are low-cost and profitable [27]. Land use improvement is one of the mitigation efforts that can be conducted to overcome drought, given its connection with the water availability, including

floods and drought [25,26]. The need to know the projected drought that will occur in the coming years will be very important, as it will provide basic data for designing mitigation activities that will be carried out to prevent the widespread impact of drought. This study aims to analyze the hydro-meteorological drought trend that occurred from 2009 to 2020 and to find out the role of land use improvement in mitigating drought, which is predicted in the next 10 years. Land use improvement carried out in this study focuses on areas according to the direction of the area functions listed in the No. SK.3000/Menhut-VII/KUH/2014 and SK.3100/Menhut-VII/KUH/2014, as well as the use of agricultural land with a slope > 25%, where these two things have not been performed to overcome the drought in the Dodokan Watershed. This research is very important and must be carried out as soon as possible, considering that drought is a very difficult threat to manage. Therefore, preventive measures are needed to formulate mitigation for handling the threat of drought in the Dodokan Watershed, and it is hoped that the information obtained from this research can support climate change mitigation activities so that the emission reduction target of 26% is achieved by 2030, as stated in the West Nusa Tenggara Governor's Regulation No. 51 of 2012.

## 2. Materials and Methods

### 2.1. Location, Tools, and Research Materials

This study was carried out in the Dodokan Watershed area, one of the priority watersheds on Lombok Island, West Nusa Tenggara, Indonesia, which has an area of 56,613.65 hectares (Figure 1).

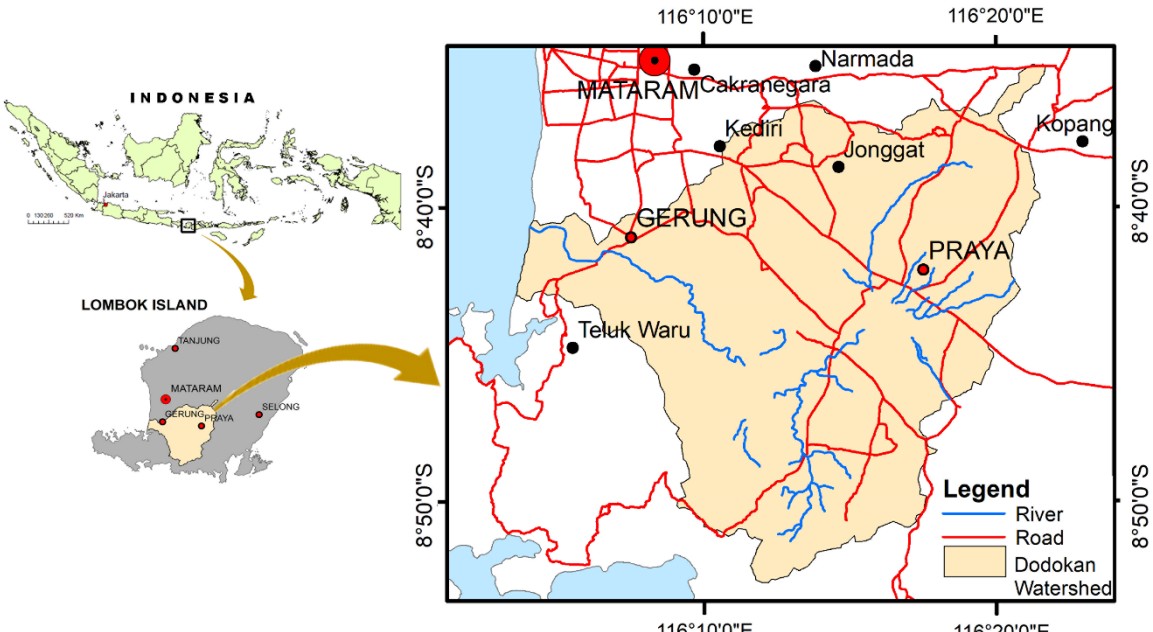

**Figure 1.** Dodokan Watershed, Lombok Island, Indonesia.

The Dodokan Watershed covers two areas, these being in the districts of Central Lombok and West Lombok. Administratively, the Dodokan Watershed covers 125 villages which are included in 15 subdistricts.

The length of the main river in the Dodokan Watershed is 31 km, while the width is 12.9 km. The Dodokan Watershed is elongated in shape with a circularity ratio of 0.36 and has a parallel dendritic flow pattern. Geologically, it is dominated by the Kalipalung formation with calcareous breccia material. The dominant soil types are the eutric cambisol and the eutric gleisol complex. Based on the analysis of rain data in 2009–2020, the climate type according to Ref. [28] is F (slightly dry) with a value of Q of 1.01.

The materials used in this study included river discharge, climatic data, population data, land use maps, and also soil maps. The tools used included stationery and laptops with software related to water balance and spatial analysis.

### 2.2. Data Collection

The river discharge and climate data used consisted of a 10-year data series originating from the Nusa Tenggara I River Basin Institute. The discharge data were obtained from the Automatic Water Level Recorder (AWLR), which was verified by direct measurement using the volumetric method. Discharge is represented as Q (m$^3$/s), which is determined by multiplying the volume V (m$^3$) by the time t (s). The discharge was either read directly from the container or estimated indirectly [29]. The data on water use was derived from the calculation of domestic and nondomestic water demands. Domestic water demands use the standard issued by UNESCO in 2002, which is 60 lt.day$^{-1}$, while nondomestic water demands are estimated to represent 15% of domestic usage as used by the Ministry of Agriculture, Settlements, and Infrastructure Territory of the Republic of Indonesia in 2000. The Water Use Index (*WUI*) is a comparison between water demand and available water supply in a place [30]. The WUI value is calculated using data on flow rates as a water supply variable and data on community water utilization as a water demand variable. Population data were provided by the Central Bureau of Statistics of Central Lombok and West Lombok District. The land use data used were derived from land use maps for 2009, 2011, 2013, 2017, and 2020, which were issued by MoEF. The soil map was obtained from the Center for Research and Development of Agricultural Land Resources year 2016.

### 2.3. Data Analysis

The description of land use was used as additional data in analyzing the potential for water availability. The description of land use was obtained by overlapping the land use maps issued by the MoEF with the boundaries of the Dodokan Watershed to obtain the type, area, and distribution of land use which would then be used in the next analysis.

The flow rate data were collected using secondary data of several years to determine the trend of the water availability and Water Use Index (*WUI*) value. The volumetric method, as applied by [29], cannot be carried out in this study because the study was conducted in the dry season, the flow conditions were very small, and the current meter could not be used. The flow rate data were then used to classify spring discharge according to Meinzer's classification system (1923) [31].

Water Use Index (*WUI*) was determined through a comparison of demand and supply of water [30]. Data on flow rates were used as a supplied variable and data on water usage were used as a demand variable [32]. *WUI* calculation employed the following formula:

$$WUI = Demands/Supplies, \tag{1}$$

The *WUI* indicator can show the potential availability of water owned by a watershed. Based on The Minister of Forestry of Republic Indonesia Regulation No. P.61/Menhut-II/2014, the *WUI* categorized as very low (*WUI* ≤ 0.25), low (0.25 < *WUI* ≤ 0.50), medium (0.50 < *WUI* ≤ 0.75), high (0.75 < *WUI* ≤ 1.00), and very high (*WUI* > 1.00). In this study, the *WUI* category was determined using the Sturgess Formula [33] as follows:

$$n = 1 + 3.322 \, log \, N \tag{2}$$

*n* is the number of categories, and *N* is the amount of data. The range value of each category is arranged based on the following formula:

$$i = \frac{Xmax - Xmin}{n} \tag{3}$$

*i* is the interval group of *WUI* category; *Xmax* is the maximum *WUI*; *Xmin* is the minimum; *n* is the number of categories.

The trend analysis was used to determine the prediction of drought that will occur in the Dodokan Watershed. The drought analysis was carried out using the Thornthwaite–Mather method [34] with the following the equation:

$$I \propto = \frac{D}{PE} \times 100 \qquad (4)$$

$$D = PE - AE \qquad (5)$$

The *Iα* is drought index (%); *D* is deficit (mm/month); *PE* is potential evapotranspiration (mm·month$^{-1}$); *AE* is actual evapotranspiration (mm·month$^{-1}$), where for the wet months (*P > PE*), the value of *AE = PE*, while for dry months (*P < PE*), *AE = P − ΔST*; *P* is precipitation (mm·month$^{-1}$); *ΔST* is change in soil moisture. The criteria for the level of dryness used are light (*Iα* = 16.77%), medium (*Iα* = 16.77–33.33%), and heavy (*Iα* ≥ 33.33%).

## 3. Results

### 3.1. The Dynamics of Land Use in the Dodokan Watershed

From year to year, land use in the Dodokan Watershed has fluctuated, as shown in Figure 2. Agricultural land (rice fields, dryland agriculture, and mixed dryland agriculture) is the dominant land use in the Dodokan watershed. The dynamics of changes that occur are not too striking compared to other land uses, except for rice fields. Primary and secondary forests experienced a major decline in 2009–2020. The total land use in the form of the forest is only about 1.98% of the 13.8% forest area that has been determined based on the Decree of the Determination of Forest Areas No. SK.3000/Menhut-VII/KUH/2014 and SK.3100/Menhut-VII/KUH/2014 issued by the Ministry of Forestry of the Republic of Indonesia, while the rest are other forms of land use such as rice fields, dryland agriculture, mixed dryland agriculture, shrubs, and even settlements. Many changes in the function of forest areas occurred in the period 2017–2020.

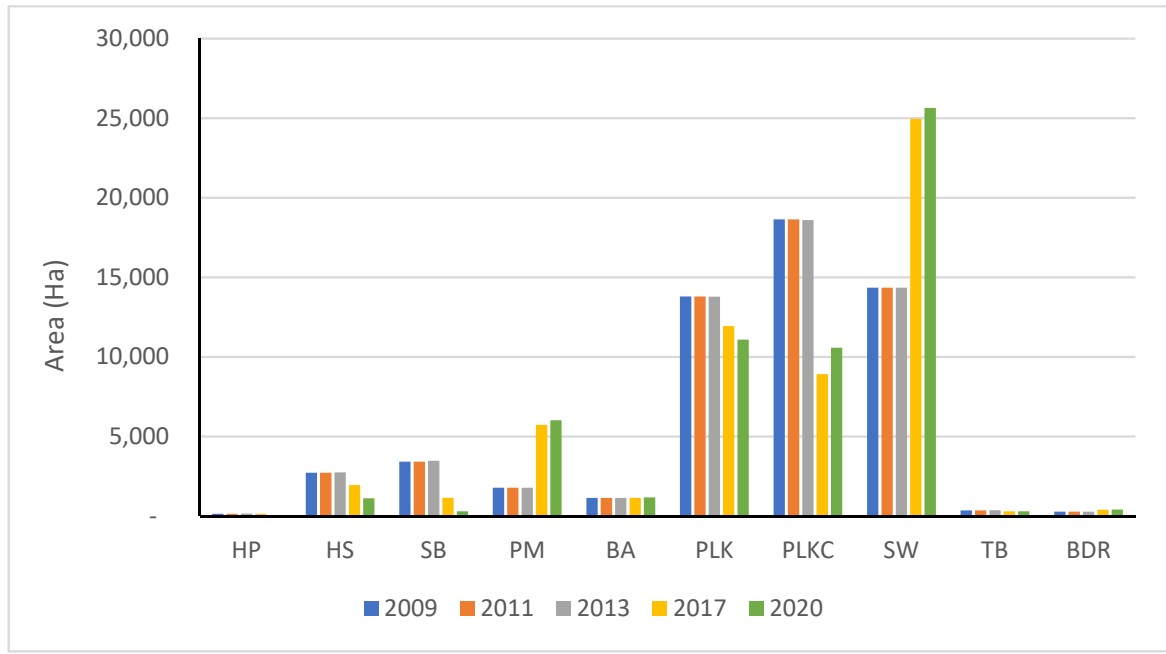

**Figure 2.** The dynamics of land use change in the Dodokan Watershed from 2009–2020. Remarks: HP: primary forest, HS: secondary forest, SB: shrub, PM: settlement, BA: water bodies, PLK: dryland farming, PLKC: mixed dryland farming, SW: rice field, TB: fishpond, BDR: airport.

### 3.2. Water Utilization

Ecologically, the Dodokan Watershed area is a producer of water sources that are utilized by the community. The community's dependence on water resources is quite high,

which in turn will have an impact on the function and sustainability of the ecosystem if it is not balanced with soil and water conservation activities. The water demand in the Dodokan Watershed during the period of 2010–2020 tended to increase (Figure 3a). The increase that occurred is in line with the increase in population every year, where based on the data from the Central Bureau of Statistics of the West Nusa Tenggara Province, population growth in NTB reaches 1% a year. By looking at this trend, it is predicted that by 2030, water consumption will have increased by 30% compared to 2010. The fulfillment of water needs depends on the water availability, one of which can be calculated through the volume discharge approach.

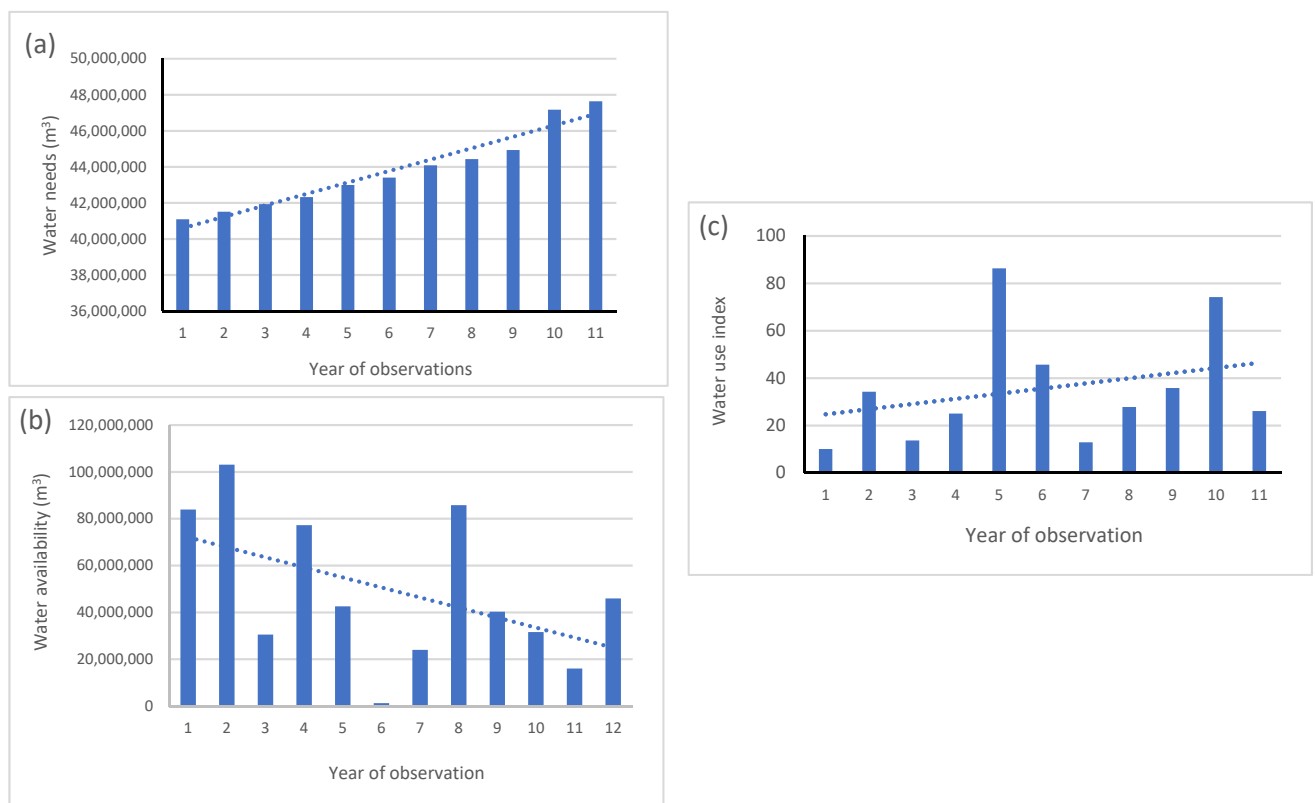

**Figure 3.** The trend of water needs in the Dodokan Watershed from 2010–2020 (**a**); water availability in the Dodokan Watershed from 2009–2020 (**b**); and the water use index in the Dodokan Watershed from 2010–2020 (**c**).

Based on data series originating from the Nusa Tenggara I River Basin Institute, namely, the AWLR Karang Makam, the water availability in the Dodokan Watershed in 2010–2020 is quite volatile every year, but tends to decrease (Figure 3b). Figure 3b shows that 2010 was the year where water availability reached the highest value, while 2014 was the lowest value of water availability in the Dodokan Watershed. Based on data of water demand and water availability, the water use index was calculated. The results show that the water use index in the Dodokan Watershed is in the very high category based on the criteria issued by the Ministry of Forestry, with the *WUI* value for 2010–2020 ranging from 10–86.3, and it tends to increase (Figure 3c). The highest *WUI* value occurred in the fifth year of observation (in 2014), while the lowest was in the first year of observation (in 2010). The results of the categorization using the Sturgess formula obtained four *WUI* categories, namely, low, medium, high, and very high. Based on the results of the analysis, there are five years in the low category (*WUI* 10–27), three years in the moderate category (*WUI* > 27–45), one year in the high category (*WUI* > 45–62), and two years in the very high category (*WUI* > 62).

*3.3. Water Balance and Drought in the Dodokan Watershed*

The results of the water balance analysis in the Dodokan watershed show that between 2009 and 2020, there were not many shifts in the months that experienced a surplus or deficit (Figure 4). In general, the water surplus occurred from December to April, while the water deficit occurred from May to November. However, a slightly different pattern occurred in 2018–2019, where the water deficiency occurred until December. Throughout the observation period, 2011 was the year with the highest number of deficit months, which was nine months.

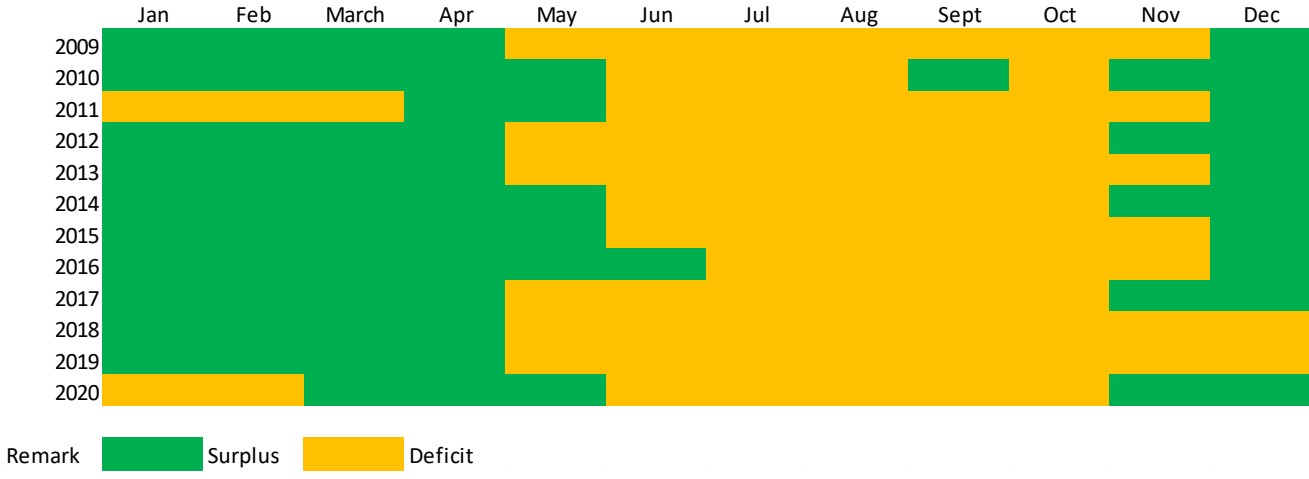

**Figure 4.** Water balance in the Dodokan watershed from 2009–2020.

The examination of the drought index revealed that the drought index value in the Dodokan Watershed during the period 2009–2020 was in the range of 1–118.4. This value is divided into three categories, namely, heavy, medium, and light (Figure 5). Figure 5 shows that the heavy drought index is mostly from June to October, while the light and medium are in May and November. A different pattern occurred in 2018 and 2019, where the drought index in December was included in the heavy category.

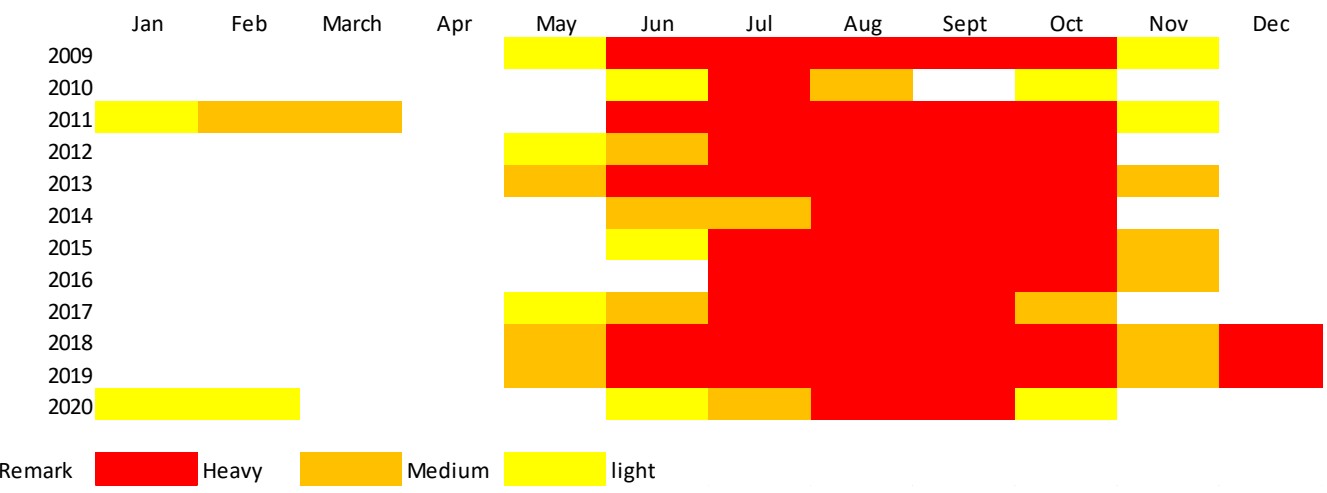

**Figure 5.** Drought index pattern in the Dodokan watershed from 2009–2020.

If it is related to the rainfall pattern that occurs, May is the month in which the rainfall begins to decline and continues to decline until it is ready to rise again in October (Figure 6). Based on observations in 2009, the trend tends to decrease.

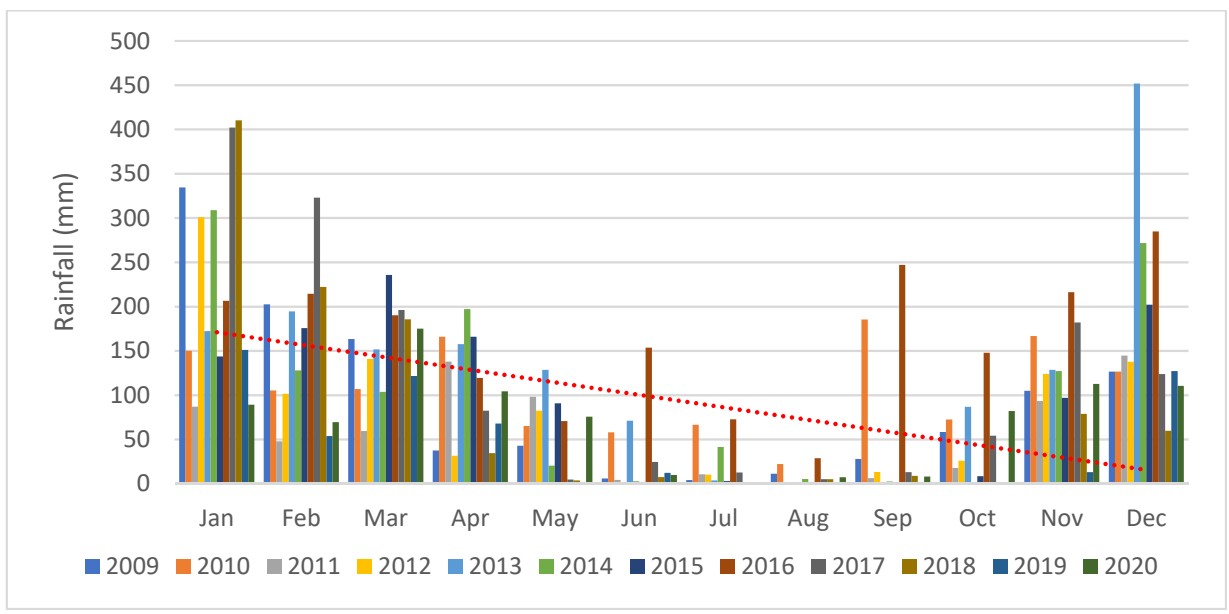

**Figure 6.** Rainfall pattern in the Dodokan watershed from 2009–2020.

The correlation between rainfall and drought index shows that rainfall significantly affects the drought index. The relationship between the two is expressed by the equation Y = −0.155x − 0.499, with a standard error of 0.023. The trend of the monthly drought index in 2009–2020 shows an increase (Figure 7).

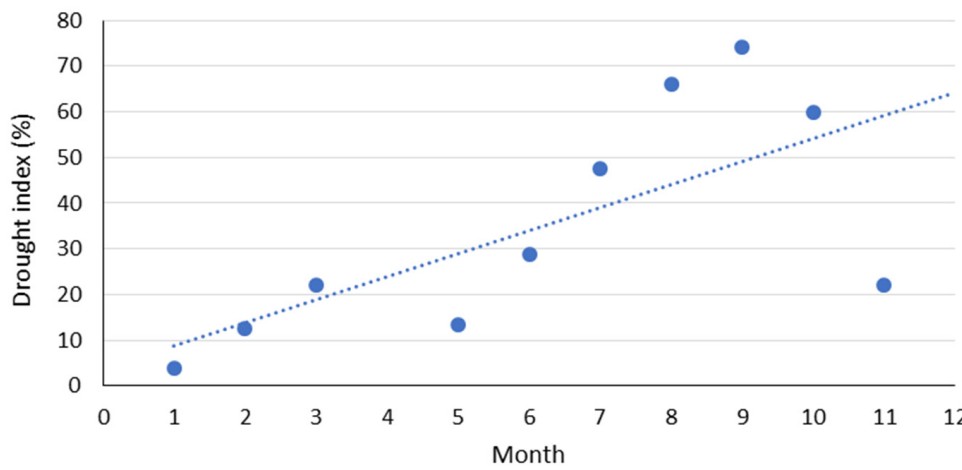

**Figure 7.** Monthly drought index trend in the Dodokan Watershed from 2009–2020.

*3.4. Drought Mitigation*

To find out what mitigation actions need to be taken to anticipate the impact of drought in the Dodokan Watershed, predictions are made for droughts that will occur in the next 5 and 10 years. This is performed in accordance with the regional action plan for climate change adaptation, which is expected to obtain results in 2030. In this study, 2020 was used as a baseline, which is considered to represent existing conditions. Two scenarios related to land use rearrangement in the Dodokan Watershed were used to determine drought mitigation measures. Furthermore, land use regulation is one of the activities that is directed to be carried out as climate change mitigation, as stated in the elaboration of the West Nusa Tenggara Governor's Regulation No. 51 of 2012.

The first scenario is to restore the forest area according to the function of the forest area in accordance with what has been determined by the Ministry of Forestry (MoF) in 2014 through the Decree on the determination of forest area No. SK.3000/Menhut-

VII/KUH/2014 and SK.3100/Menhut-VII/KUH/2014. The second scenario is to rearrange the use of agricultural cultivation land on land with a slope of >25% in accordance with the Regulation of the MoF No. P.61/Menhut-II/2014. In this regulation, it is stated that to produce a good watershed function, cultivated land on land with a slope of more than 25% must not exceed 30%.

Based on the regional function directives issued by the MoF in 2014, the forest area in the Dodokan Watershed is 7024.83 Ha, or 13.85% of the total area of the Dodokan Watershed, which consists of 1724.2 Ha of protected forest and 6114.8 Ha of production forest. In the existing condition, the total forest area in the Dodokan Watershed is only 1122.8 Ha, consisting of primary and secondary forest. Scenario one is made to restore the function of the forest area, with the assumption that within 5 years, 50% of the forest area will be realized according to the direction of the area function, while in the period of 10 years, the entire forest area will be realized as forest according to the existing directions.

The results obtained from the first simulation scenario show that there will be an increase in the amount of surplus water in 2025 and 2030, as illustrated in Figure 8. The simulation results show that in 2025 and 2030, the surplus of water occurs within four months, namely, from January to April, while in 2020, an existing condition only occurs for one month, namely, April. Meanwhile, under deficit conditions, in 2030 it appears to decrease from 2025 to 2030.

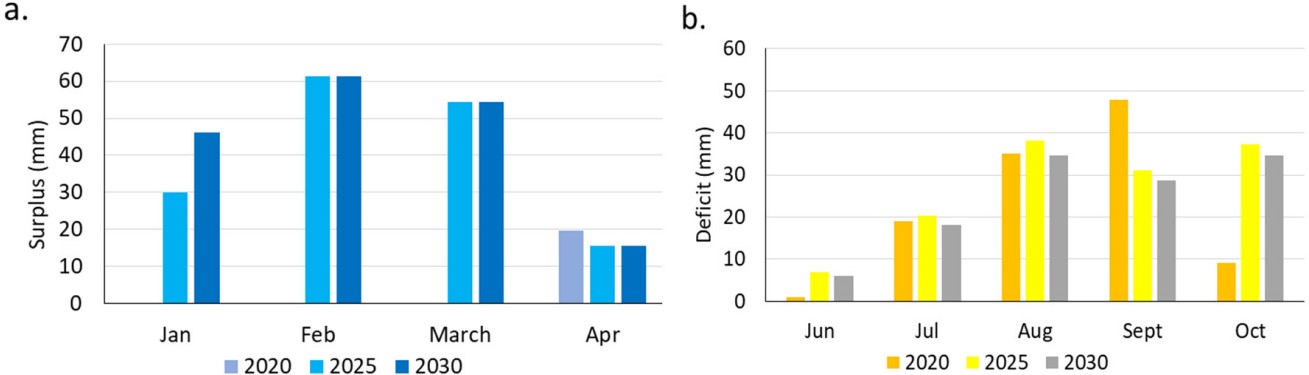

**Figure 8.** Changes in the water surplus (**a**) and deficit (**b**) in the years 2020, 2025, and 2030 based on the simulation with scenario one in the Dodokan Watershed.

A shift in the criteria for the drought index also occurred in 2025 and 2030 in the simulation, as can be seen in Figure 9. This indicates that the application of scenario one is quite effective as a mitigation effort against drought in the Dodokan Watershed.

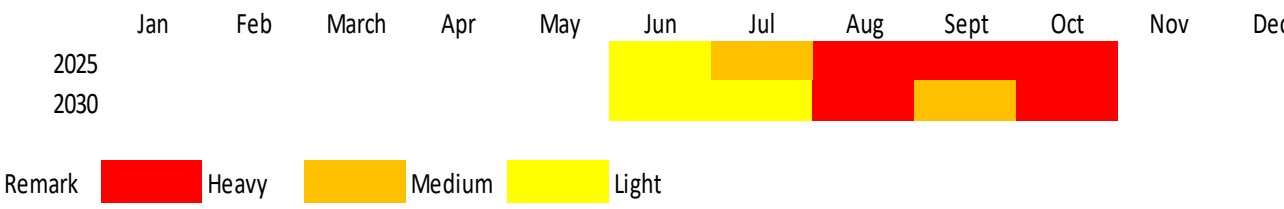

**Figure 9.** Drought criteria pattern in 2025 and 2030 based on the simulation with scenario one in the Dodokan Watershed.

In addition to experiencing a shift in the drought index criteria as illustrated in Figure 9, the drought index value in the Dodokan Watershed also changed from the existing conditions (in 2020), as shown in Figure 10. In 2025, the decline in drought index occurred in January, February, July, August, and September, while the increase occurred in June, August, and October. At the end of the experiment (in 2030), the drought index value decreased in January, February, July, and September, while it increased in June, August,

and October. The drought index's value changed very little in February, but it increased significantly in October.

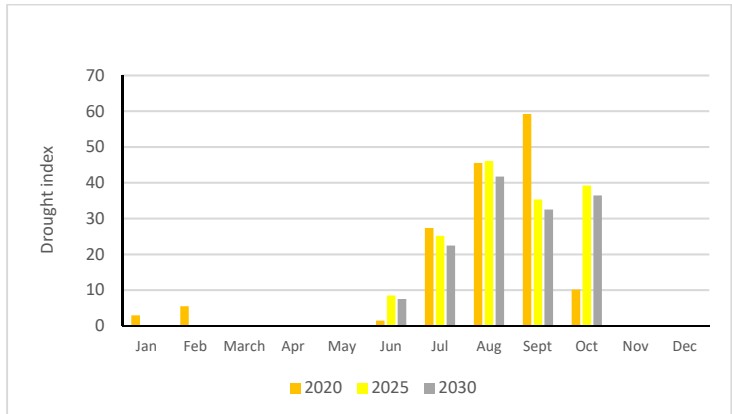

**Figure 10.** Change of the Drought Index in the years 2020, 2025, and 2030 based on the simulation with scenario one in the Dodokan Watershed.

In the second scenario, land use change focuses on agricultural cultivation in the form of dryland agriculture (PLK), mixed dryland agriculture (PLKC), and rice fields on land with a slope of more than 25%. In the existing condition, the percentage of agricultural land is 80.1%, while after the simulation it becomes 23.5% (Table 1). The simulation results of scenario two show that there is an additional month of surplus water from the existing condition, which was only one month, namely, April, to four months, namely, from January to April, but in terms of quantity, the amount of surplus decreased (Figure 11a). This condition is different from the water deficit, where in Figure 11b it can be seen that the total water deficit tends to increase.

**Table 1.** Comparison of land use area on slopes >25% in existing conditions and after simulation using scenario two.

| Land Use | Area (ha) | | | |
|---|---|---|---|---|
| | Existing | % | Simulation | % |
| Primary forest (HP) | 183 | 11.4 | 268 | 16.6 |
| Secondary forest (HS) | 0 | 0.0 | 856 | 53.0 |
| Shrub (SB) | 39 | 2.4 | 13 | 0.8 |
| Settlement (PM) | 97 | 6.0 | 97 | 6.0 |
| Water bodies (BA) | 1 | 0.1 | 1 | 0.1 |
| Dry farming (PLK) | 169 | 10.5 | 9 | 0.6 |
| Mix dry farming (PLKC) | 985 | 61.1 | 236 | 14.6 |
| Paddy land (SW) | 139 | 8.6 | 134 | 8.3 |

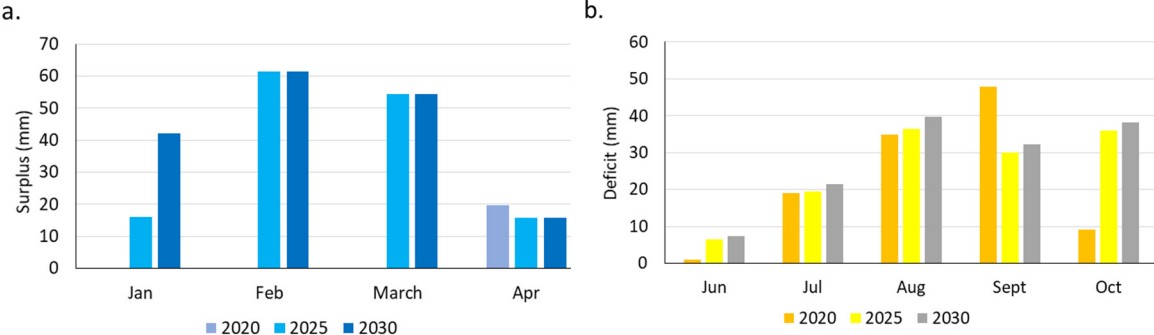

**Figure 11.** Changes in the water surplus (**a**) and deficit (**b**) in 2025 and 2030 based on the simulation with scenario two in the Dodokan Watershed.

As the result of the water deficit analysis, the value of the drought index in the Dodokan Watershed by using scenario two also experienced an increase in the value of the drought index (Figure 12). This condition further strengthens the fact that scenario two is not sufficient to overcome the drought in the Dodokan Watershed.

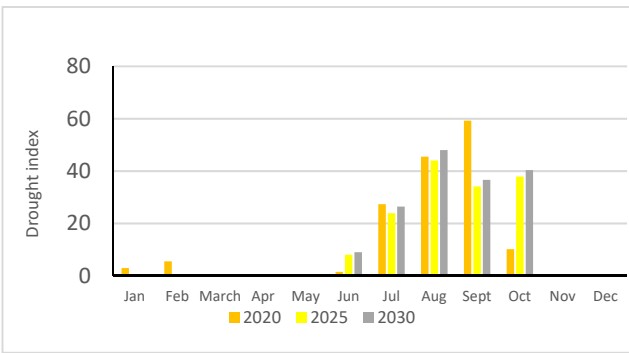

**Figure 12.** Change of the Drought Index in the years 2020, 2025, and 2030 based on the simulation with scenario two in the Dodokan Watershed.

## 4. Discussion

According to Law Number 17 of 2019 concerning Water Resources, water resources are determined as water, water sources, and the water resources contained therein. Water resources are water-based resources that are useful or potentially helpful to humans. The availability of water is the amount of water available so that it can be used for various purposes [35]. Water availability can include quantity, quality, as well as energy or water pressure. If water is not widely available, people will tend to be frugal and use water wisely. Water demand is the amount of water used for various purposes or community activities in the area. People use water for both domestic and non-domestic purposes, such as water needs for households and public facilities, offices, education, irrigation, animal husbandry, industry, fishing, and others. Domestic water needs are those that are met in private residential areas on a daily basis. Domestic or residential water demands include drinking, cooking, showering, cleaning toilets, washing vehicles, irrigating plants, and so on [36]. Nondomestic water needs include commercial use, institutional needs, and industrial needs.

The increasing water demand in the Dodokan Watershed, as shown in Figure 3a, will pose a threat to the fulfillment of water needs, especially if the availability of water is not sufficient as stated by [10]. The trend of decreasing water availability, as shown in Figure 3b, is a condition that needs to be taken in mind in line with the increasing demand for water, as shown in Figure 3a. The water availability is indicated by a continuous flow. Rainfall is one input factor that affects it. When viewed from the average rainfall area in the Dodokan Watershed, in 2010 and 2014 the values were almost the same, at 1292 mm and 1211 mm, respectively. Thus, rainfall is not the main factor affecting the water availability in the Dodokan Watershed. Ref. [37] stated that the flow rate is also influenced by the proportion of rainfall that flows into a surface runoff, infiltration of rainwater into the soil, land cover, and the application of soil and water conservation techniques. Land use changes in the Dodokan Watershed in 2009–2020 occurred dynamically, especially forest land conversion, which is thought to be a factor that affects the water availability in the Dodokan Watershed, as stated by [8]. Forest land conversion into other forms of land use must be stopped immediately to restore the hydrological role of the watershed, as stated by [38]. The analysis of *WUI* as shown in Figure 3c means that the water demand in the Dodokan Watershed has exceeded the water availability, so it is necessary to have other water sources than those from the Dodokan Watershed, so that the water needs of the community can be fulfilled. By using the categories issued by the Ministry of Forestry, the condition of the Dodokan Watershed becomes very vulnerable to drought because *WUI* has very high criteria. Meanwhile, by using the categories that have been compiled in this

study, the *WUI* criteria that occurred in the Dodokan Watershed appear to be very diverse, with the highest *WUI* criteria occurring in 2014 and 2019 (Figure 3c). This is in accordance with the existing condition of water availability, where in both years the water availability was at the lowest level compared to other years (Figure 3b).

The water balance of the Dodokan Watershed, as shown in Figure 4, also occurs in various other areas on Lombok Island, wherein 2011 was categorized as the worst drought as stated by [39]. This is possibly influenced by the presence of El Niño, which causes a shift at the beginning of the rainy season, as stated by [4]. Figure 4 shows that the water balance pattern in 2011 is almost the same as in 2020. This is presumably due to the same rainfall pattern in both, as stated by [40,41]. Figure 4 shows that the water deficit has increased in frequency and duration. The lowest number of months of deficit occurred in 2016, which was four months, while the highest occurred in 2011, which was nine months. It will affect efforts to meet water needs [42]. Figure 3b shows a downward trend in water availability in the Dodokan watershed, which is the opposite of the water deficit in Figure 4. This condition strengthens the argument that water deficits are related to water availability. The category of drought index in December 2018 and 2019 (Figure 5) was not only caused by a lack of rainfall in that year, but it was also thought to be due to an anomaly of wind vectors that caused drought as stated by [43]. The rainfall pattern shown in Figure 6 shows that in the May–October period there was a decrease in rainfall. During this period, most of Indonesia, including Lombok, was experiencing a dry season [44]. Some areas in Indonesia also experience drought in these months [45]. The upward trend in the monthly drought index as shown in Figure 7 is thought to be caused by the global warming phenomenon that affects water scarcity as mentioned by [22]. By looking at this trend, the drought index in the Dodokan Watershed for the next 10 years can be predicted using various assumptions related to the determination of the water balance, including rainfall, air temperature, and soil moisture reserves. Prediction is carried out by using trial and error. The predictions made will be useful for determining the mitigation actions that will be implemented to overcome the drought in the Dodokan Watershed.

Land use improvement is carried out in Dodokan Watershed because various studies show that land use changes have considerable impact on water availability [46–48]. Restoring the function of the forest (scenario one) was chosen as an alternative because the role of the forest is very important in regulating the water system [24,38]. Land with a large slope used for agriculture will have an impact on erosion and landslides which will indirectly affect the environmental balance, including the water cycle [49], so that the land use arrangement on land with a large slope is chosen as the second scenario. The simulation with the first scenario shows an increase in the number of months of surplus, while for months the deficit remains constant but there is a decrease in the value of the deficit (Figure 8). Scenario one is also able to reduce the number of months that have a drought index in the heavy category (Figure 9) and gradually reduce the value of the drought index in 2030 (Figure 10). This condition explains that the forest has a role in storing water, as stated by [38]. Figure 10 shows that, in comparison to the existing situation (in 2020), the value of the drought index in October increases significantly in 2025 and 2030. Commonly, the climax of the dry season on Lombok Island occurred in October, as depicted in Figure 6, whereas the average annual rainfall is on the decline. The availability of water is expected to decline due to the trend of declining precipitation and rising population, which would increase the value of the drought index as indicated in Figure 6, which occurred in 2025 and 2030. This is consistent with the assertion made by [21,22] that there is a connection between drought and water availability.

In the second scenario, agricultural land (PLK, PLKC, and SW) on a slope of >25% must be returned according to its function, most of which is forest area. The simulation shows that a 56.6% reduction in agricultural land area can be used as an effort to mitigate drought (Table 1). Change in agricultural land is assumed by converting the agricultural land used in the slope above 25% to primary or secondary forest land with an agroforestry pattern. It is hoped that this pattern will not only benefit the ecology but also the local

economy [50]. The simulation in this scenario shows that although it is able to increase the amount of surplus water, it has not been able to reduce the water deficit (Figure 11) and also increase the value of the drought index (Figure 12). The increase in the water surplus means that changes in vegetation density will affect the water balance as described in [24]. However, land use changes focused on land with a slope above 25% are not good enough to reduce the water deficit, but are good enough to increase the water surplus in the Dodokan Watershed.

## 5. Conclusions

Drought in the Dodokan Watershed has a tendency to increase from year to year. Simulation using land use change scenarios as one of the mitigation actions is needed to minimize the risks that will occur. The scenario of land use improvement by restoring forest area according to the direction of area functions that have been implemented by the government is a more effective way than the scenario that improves agricultural land use on a slope above 25%. Changes in land use relating to owned land and state land are sensitive matters for the community, so it is necessary to conduct socialization related to the function of forest areas. One of the important things that needs to be carried out as a follow-up to this research is the existence of strict law enforcement so that changes in utilization by returning forest areas according to the direction of the area functions listed in the No. SK.3000/Menhut-VII/KUH/2014 and SK.3100/Menhut-VII/KUH/2014 can run well, so that drought reduction can be achieved by 2030.

In Indonesia, every local government authority has a planning document in the form of land use directions as outlined in the Regional Spatial Plan (RTRW), which has legal force and becomes a reference for every land use in the region. The RTRW document is prepared periodically in a participatory manner by involving stakeholders through development planning deliberation activities (Musrenbang), and will be evaluated periodically. The results of the recommendations obtained in this study have the opportunity to improve the planning documents that have been prepared, where the act of restoring forest areas in accordance with the direction of the functions that have been determined is the most effective step taken to prevent droughts that occur in the Dodokan Watershed. The results of this study need to be used as the basis for preparing plans for soil and water conservation activities in the Dodokan Watershed, as well as for encouraging the acceleration of climate change mitigation efforts as mandated in the West Nusa Tenggara Governor's Regulation No. 51 of 2012 on climate change mitigation action plans in West Nusa Tenggara Province.

**Author Contributions:** Both authors (R.N. and A.K.) played an equal role as key contributors, discussing the fundamental concepts and outline equally, providing critical input on each part, and assisting in the shaping and writing of the work. The published version of the manuscript has been read and approved by all authors. All authors have read and agreed to the published version of the manuscript.

**Funding:** This research was funded by Universitas Gadjah Mada through the Postdoctorate Scheme Research Fund, based on the Decree Number: 6282/UN1.P.III/SK/HUKOR/2021 concerning Universitas Gadjah Mada Postdoctoral Program Recipients in 2021 and Assignment Letter Number: 6144/UN1.P.III/DIT-LIT/PT/2021 regarding Universitas Gadjah Mada University Postdoctoral Program Recipients for Fiscal Year 2021.

**Institutional Review Board Statement:** Not applicable.

**Informed Consent Statement:** Not applicable.

**Data Availability Statement:** Not applicable.

**Acknowledgments:** We would like to thank Universitas Gadjah Mada (UGM) for the support of this research through the Postdoctorate Scheme Research Fund. Special thanks to UGM Research Directorate and UGM Reputation Improvement Team towards World Class University—UGM Quality Assurance Office, for facilitating this program, 2021.

**Conflicts of Interest:** The authors declare no conflict of interest. The founding sponsors had no role in the design of the study; in the collection, analyses, or interpretation of data; in the writing of the manuscript; or in the decision to publish the results.

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
