# Peer review of "Land Use Improvement as the Drought Mitigation to Manage Climate Change in the Dodokan Watershed, Lombok, Indonesia"

_land, doi:10.3390/land11071060_

Round 1
Reviewer 1 Report
The presented work concerns land use improvement as the drought mitigation to manage climate change in the Dodokan Watershed, Lombok, Indonesia. The issue described in the paper is important and it is good that the authors have undertaken such research. The work is written in an appropriate way, includes an introduction, methodology, results, discussion and conclusions. The authors used the latest literature, appropriately selected to the topic of the work. My comments regarding the work concern: 1. The abstract should be a summary of the research problem, it should contain the methods used, main data and conclusions. Please edit the text. 2. In the introduction, please indicate the reasons for undertaking the research and indicate its novelty. 3. Fig. 1 - bad quality. 4. Figures 3 a and c give the years as numbers (1 to 11) and figure 3 b as years (2009 - 2020), isn't this a mistake? 5. Minor editorial errors are marked in yellow in the text.

Reviewer 2 Report
The paper titled “Land use improvement as the drought mitigation to manage climate change in the Dodokan Watershed, Lombok, Indonesia” is rather interesting and well-organized but needs some minor revisions. Here are some detailed comments:
L30: Please add reference: “Drought is a problems that occurs frequently in many areas of the Lombok Island”.
L30: Please review the manuscript to improve it grammatically. “a problem”
L31: Please quantitatively state what low rainfall and the short number of rainy days mean here.
L55: It would be great if you can discuss changes in water shortage characteristics as well to show how much the meteorological and hydrological drought can affect the gap between water supply and water demand. Please add references accordingly.
L95: Please highlight what type of drought you are investigating.
L 169: Please add references for “Water Utilization Index (WUI)”.
L 176: it would be recommended to also categorize the WUI>1.00 as (e.g., WUI>1.00, WUI>2.00, etc.,) to figure out how the land use improvement can attenuate both minor and major water deficits with different magnitude.
L 384: The figures are not clear. The information on the figure should be independent of the manuscript’s texts. For example, the author should highlight that if all 2020, 2025, and 2030 results are after the implementation of the mitigation scenario. Please add a similar figure for baseline conditions. Additionally, discuss the reason for the high jump from 2025 to 2030 in October.
L 390: Please add the quantitative values here. I recommend adding a table to compare the baseline and two scenarios.
L 506: The author is recommended to explain how their finding can help decision-makers, and land and water planners in the future including the current and potential limitations.
